# Complement as a Therapeutic Target in Systemic Autoimmune Diseases

**DOI:** 10.3390/cells10010148

**Published:** 2021-01-13

**Authors:** María Galindo-Izquierdo, José Luis Pablos Alvarez

**Affiliations:** Servicio de Reumatología, Instituto de Investigación 12 de Octubre, Universidad Complutense de Madrid, 28040 Madrid, Spain; jlpablos@h12o.es

**Keywords:** complement system, pathogenesis, therapeutic blockade, rheumatic autoimmune diseases

## Abstract

The complement system (CS) includes more than 50 proteins and its main function is to recognize and protect against foreign or damaged molecular components. Other homeostatic functions of CS are the elimination of apoptotic debris, neurological development, and the control of adaptive immune responses. Pathological activation plays prominent roles in the pathogenesis of most autoimmune diseases such as systemic lupus erythematosus, antiphospholipid syndrome, rheumatoid arthritis, dermatomyositis, and ANCA-associated vasculitis. In this review, we will review the main rheumatologic autoimmune processes in which complement plays a pathogenic role and its potential relevance as a therapeutic target.

## 1. Introduction

The complement system (CS) includes more than 50 proteins that can be found as soluble forms, anchored to the cell membranes, or intracellularly [1]. Most of the complement proteins are synthesized in the liver, although other cell types, especially monocytes/macrophages, can produce them. Tissue distribution is variable and a higher concentration of these proteins is found in certain locations such as the kidney or brain. The main function of the CS is to recognize and protect against foreign or damaged molecular components, directly as microorganisms, and indirectly as immune complexes (IC). This is achieved through different mechanisms such as opsonization and phagocytosis, direct cell lysis, and triggering of pro-inflammatory responses by anaphylotoxins. Other homeostatic functions of CS are the elimination of apoptotic debris, neurological development, and the control of adaptive immune responses [2].

The activation of CS occurs through three main pathways—classical, lectin, and alternative—that converge in C3 activation (Figure 1). Each pathway is activated by different conditions, but all three pathways result in the creation of a pro-inflammatory environment, the deposition of large amounts of C3 in target cells (opsonization), and membrane disturbance, including lysis by the membrane attack complex (MAC). The classical pathway is activated by the binding of C1q to the Fc portion of immunoglobulin G or M in the IC [3]. Upon binding the target surface, C1q undergoes a structural change with activation of C1r, which subsequently divides and activates the two C1s molecules with serine protease activity [4,5]. Active C1s are divided into C4 and C2 to generate the C3 convertase, C4b2a. Once C3 is activated, the larger fragment C3b can covalently bind to the target surface or to C4b in the C4b2a complex. This last reaction generates the C5 convertase C4b3b2a, and the terminal pathway. Once C3b is deposited on a surface, the alternative pathway can be activated forming the C3b-FactorB complex, which is also activated, giving rise to C3bBb convertase by the action of Factor D. The lectin complement pathway has an activation scheme comparable to that of the classical pathway, but lectins (carbohydrate-linked proteins) replace antibodies and lectin-associated proteases replace C1r and C1s [6,7]. The lectin-associated serine proteases (mannan-associated lectin-binding serine proteases, MASPs) bind to mannose and cleave C4 and C2 factors [8]. The alternative pathway does not require antibodies or contact with a microbe to be activated [9,10]. Instead, C3 is constantly self-activated (C3 tick-over) at a low level, a process that is rapidly amplified in the presence of a microbe, a damaged host cell, or importantly, by deficiency of complement regulatory proteins. The deposition of C3b on a target can be efficiently amplified by the feedback loop of the alternative pathway.

All these pathways result is the activation of inflammatory responses by releasing pro-inflammatory peptides known as anaphylatoxins (C3a, C4a, C5a), due to their ability to induce mast cell and basophil degranulation and hence the release of vasoactive and chemoattractant mediators [11]. Cytokine signaling contributes to an up-regulation of anaphylatoxin receptors (C3aR, C5aR) by endothelial cells in small vessels and circulating leukocytes. Binding of C3a and C5a to the reciprocal receptors on these cells enhances the release of cytokines and eicosanoids that contribute to an increase in vascular permeability, vasodilation, and leukocyte extravasation. Anaphylatoxins up-regulate adhesion molecules on endothelial cells and leukocytes, facilitating the adhesion of leukocytes to the vascular wall and their subsequent transmigration into the interstitial tissue at sites of inflammation. C3a and C5a stimulate mast cells to release histamine and proteases that also contribute to vascular alterations. Monocyte-derived macrophages are among the first cells encountering non-self-antigens, and through the pattern recognition system they initiate an innate immunity response [12]. While C3a and C5a are well recognized anaphylatoxins with specific receptors, C4a has also been found to be a ligand for protease-activated receptors 1 and 4 (PAR1 and PAR4), also with effects on endothelial permeability and inflammatory cell recruitment.

There is a complex system of control of CS activation (Figure 1). The regulatory proteins inhibit the CS by destabilizing the activation complexes or by mediating the specific proteolysis of activated fragments. Complement pathways are regulated in the following critical steps: activation or initiation, amplification and convertases formation, or MAC-dependent lysis [1,13,14]. In the classical pathway, the C1 inhibitor (C1Inh), a serinprotease of the “serpin” family, binds to each C1r and C1s sub-components of the C1 complex. This causes its dissociation and release of C1q, which binds to the Fc portion of immunoglobulin G and M [15,16]. C1Inh performs a similar function in the lectin pathway. The activity of C3 convertases is regulated by a family of complement binding proteins. These include membrane proteins such as the decay acceleration factor (DAF or CD55), membrane cofactor protein (MCP or CD46), and the CUB and Sushi multiple domains 1 (CSMD1) protein [13,17]; in addition to plasma proteins such as the C4b-binding protein (C4BP) that regulates the classical and lectin pathway conversion containing C4b and C4b, factor H that regulates C3b, alternative pathway conversions containing C3b and C5 fragments containing either one C3b (from the classical/lectin pathway) or two C3b (alternative pathway).

The S protein (also known as vitronectin) controls the soluble MAC by binding to the C5b-7 complex, thus preventing its anchorage to cell membranes [18]. The MAC that is deposited in tissues is inhibited by the CD59 [19]. This widely expressed glycolipid-anchored membrane protein has binding sites for both C8 and C9 and therefore inhibits the final steps of MAC assembly.

Finally, receptors for complement activation fragments are expressed on many host cells, including peripheral blood, endothelial, and epithelial cells [15,20,21]. Receptors for C4b and C3b are present on most cells of the immune system and promote pathogen destruction and the generation of the adaptive immune response. The receptors for C3a and C5a are widely distributed where they trigger the local inflammatory response (innate immunity) and also cell activation to prepare the adaptive immune response. Together, these receptors promote the adhesion and phagocytosis of microorganisms and IC.

The CS plays an important physiological function in defense but its pathological activation plays prominent roles in the pathogenesis of most autoimmune diseases including rheumatoid arthritis (RA), multiple sclerosis (MS), myasthenia gravis (MG), systemic lupus erythematosus (SLE), ANCA-associated vasculitis (AAV), Sjögren’s syndrome (SS), antiphospholipid syndrome (APS), dermatomyositis, systemic sclerosis, C3 glomerulopathies, and many other diseases [22].

In this review, we will review the main rheumatologic autoimmune processes in which complement plays a pathogenic role and therefore its potential relevance as a therapeutic target.

## 2. Systemic Lupus Erythematosus

The pathogenic involvement of CS in SLE has two different facets. On the one hand, classical CS activation significantly contributes to inflammatory tissue damage. On the other hand, genetic deficiencies of complement factors are associated with an increased risk of developing SLE. Deficiency of early complement factors is associated with a higher risk of developing SLE. However, the risk varies depending on the factor affected by the genetic deficiency, C1q (90–93%), C1r/C1s (50–57%), C4 (75%), and C2 (10%). Several mechanisms may be involved in the high risk of SLE in C1 deficiency. C1q is an important factor in the recognition and clearance of apoptotic cells and circulating IC that seem potent stimuli of the innate and adaptative responses in SLE. Furthermore, it has a regulatory role by restraining CD8 T-cell responses that result in autoimmunity in animal models [23,24]. Successful treatment of C1q-deficient patients in SLE with fresh frozen plasma or hematopoietic stem cell transplantation has been reported [25]. C2 deficiency, but not C1q or C4 deficiency, is able to circumvent the mechanisms modulating complement activation. In a registry of complement deficient patients, a third of C2 deficient developed a SLE-like disease [22,26]. Interestingly, in the very rare cases of homozygous deficiency of C3, the most critical protein where all three pathways of complement activation converge, there is no association with SLE [27]. Complete deficiency of C4a and C4b is rare but strongly associated with SLE. Partial defects in C4 factors are common in SLE, and are part of the MHC haplotypes associated with an increased SLE risk, as well as to other multigenic autoimmune diseases such as diabetes mellitus type 1 or primary biliary cirrhosis [28]. Genetic deficiency of MBL has also been associated with the development of autoimmunity, including SLE and other types of arthritis [29,30,31]. Mutations in factors I and H have been linked with an increased risk for SLE, atypical hemolytic uremic syndrome (aHUS), paroxysmal nocturnal hemoglobinuria (PNH), and C3 glomerulopathies [32]. SLE linked to genetic complement deficiencies has certain clinical peculiarities such as earlier onset and more frequent development of photosensitive rash and neurological involvement but less common development of pericarditis, and renal or pulmonary involvement.

CS activation is one of the most important mediators of inflammatory tissue injury induced by autoantibodies and IC in SLE. The effector mechanisms of CS activation in SLE involve cytotoxicity, inflammatory infiltration by increased endothelial permeability, and leukocytes chemoattraction. CS also contributes to class I interferon production by dendritic cells, and increases T and B cell autoimmune responses in SLE.

The pathogenic activation of CS in human SLE was early described [33,34], including low total hemolytic activity of complement (CH50) and decreased levels of C3 and C4, particularly in patients with most severe manifestations such as active nephritis [35]. Complement and Ig deposition in involved tissues and cells of SLE patients such as collocation of IgG, IgA, and IgM with C1q, C4, and C3 factors (and C5b-9; the so-called “full house” pattern) in the glomeruli is almost exclusively present in patients with lupus nephritis [36]. Complement split products such as C3d and C5b-9 can also be detected in the urine of SLE patients [37].

Complement components such as C1q, MBL, C3, and ficolin-3 may also be target antigens for autoantibody responses. Anti-C1q antibodies are often detected in SLE as well as other autoimmune diseases including hypocomplementemic urticarial vasculitis, and with less prevalence in systemic sclerosis, RA, undifferentiated connective tissue disease, and SS [38,39,40,41]. In SLE patients, anti-C1q antibodies are associated with proliferative lupus nephritis (LN), and their level correlates with the degree of activity [42,43]. Autoantibodies recognizing C3b have also been described in patients with LN [44,45] and correlate with plasma levels of C4 and C3 [46,47]. Moreover, anti-C3b levels correlate with those of anti-DNA [48], suggesting the hypothesis that the double presence of anti-DNA and anti-C1q Ab may activate the complement system inducing the appearance of anti-C3b Ab [47]. Autoantibodies against soluble regulators such as anti-C1-INH autoantibodies have also been described in patients with SLE associated with high disease activity scores [49].

A large variety of functional CS abnormalities have been described in SLE. The complement receptor 1 (CR1) binds to targets opsonized by C4b and C3b, and is involved in the transport and clearance of IC. Reduced CR1 levels have been observed in SLE, therefore contributing to IC mediated pathology [50]. Reduced levels of CD55 and CD59 have also been found in SLE lymphocytes, possibly contributing to the development of lymphopenia [51,52]. Higher soluble CD46 levels have been described in patients with active SLE compared to healthy controls or patients with other autoimmune diseases [53]. Serum levels of other CS regulators such as Factor I also correlate negatively with the degree of disease activity [54]. Finally, increased levels of ficolin-1 and -3 have been found in SLE patients with a positive correlation with accrual damage and arterial thrombosis [55].

Additional effector mechanisms have also been described in SLE. Complement fragments can activate neutrophils and platelets, inducing NETosis and favoring platelet aggregation and adhesion to the vascular endothelium [56,57]. SLE patients show increased levels of C1q, C3d, and C4d in their platelets, especially in patients with a history of venous thrombosis. One study also suggests that small dense particles of HDL may activate the CS being involved in subclinical atherosclerosis in SLE patients [58].

## 3. Antiphospholipid Syndrome

Different animal models have shown the role of complement in the development of thrombosis and obstetric pathology associated with antiphospholipid antibodies (APL) [59,60] (Table 1). Animal models have shown that complement blockade or deficiency prevents the development of thrombosis or fetal loss induced by the transfer of APL [61,62]. The disease requires an intact alternative pathway, and treatment of mice with an inhibitory antibody to factor B protected them from fetal loss [63]. The use of a C5a receptor antagonist was also protective in this model. In animal models, specific blockade of the C5b-9 complex or the C3 and C5 genetic deficiency protects from the development of thrombosis [64,65].

The suspected involvement of complement in the APS is well known, with evidence of increased circulation of soluble C5b-9 (sC5b-9) [66], Bb, and C3a [67]. Co-location of the complement with the endothelial β2GP1 further supports this association [68]. The antigen β2GP1 contributes to the regulation of complement and coagulation, and antibodies against it are probably the main pathophysiological driver of APS [69,70]. Activated complement fragments can induce the activation of endothelial cells and a prothrombotic phenotype through the MAC or C5a receptor (CD88). Studies in mice have suggested that fetal regulation of the complement system is essential for embryo and fetal survival [60]. Blocking the complement pathway at the level of C3 activation and monoclonal anti-C5 antibody prevented fetal loss and thrombophilia in mouse models, respectively [59,65].

Resistance to endothelial cell activation and thrombosis was also found in C3- and C5-deficient mice. It is also postulated that heparin, by inhibiting complement activation, may play a role in reducing miscarriages in APS [71].

Indirect evidence also suggests that in patients with APS, CS also plays a role in thrombosis and obstetric pathology. CS activation and hypocomplementemia have been observed in patients with primary APS, but not a temporal correlation with thrombosis [72].

## 4. Sjögren’s Syndrome

The presence of C4A null alleles are part of the immunogenetic risk of SS, as they are in SLE, and consistently, in SS, low serum C4 levels are often observed [73]. Decreased levels of C4BP have been shown to correlate with increased disease activity [74]. Anti-C1q antibodies have also been detected in some patients with SS [38]. Hypocomplementemia in SS is associated with extraglandular systemic manifestations such as fever, joint or skin diseases, vasculitis and peripheral neuropathy, and with some immunological phenomena (cryoglobulinemia and positive rheumatoid factor). Moreover, its presence is associated with a worse prognosis with a higher risk of lymphoma and higher mortality [75]. However, longitudinal studies are lacking, and therefore a temporal correlation between hypocomplementemia and some of these manifestations cannot be established. Therefore, the pathogenic meaning of CS activation in SS is unknown, exception taken of the vascular pathology caused by cryoglobulins. The clinical value of determination of complement factors in the monitoring of these patients is also uncertain.

## 5. Rheumatoid Arthritis

Multiple evidences suggest that excessive local CS activation in the joints is a relevant factor in the pathogenesis of arthritis in RA. Synovial tissue seems a complement factor rich environment, since synovial fibroblasts, macrophages, and endothelial cells produce complement factors in the joints [76,77,78]. Complement cleavage products such as IC containing C3a, C3c, C5a, sC5-9, Bb, C1-C1INH are increased in synovial fluid from RA patients [79,80,81,82,83,84,85,86,87,88,89]. Other signs of complement activation, such as C1q-C4 complexes, are present in the circulation of RA patients [90], but systemic consumption of complement factors is not present in RA as in other systemic autoimmune diseases. Instead, increased plasma levels of C3 and C4 in active RA patients are observed, reflecting increased hepatic synthesis induced by IL-6 mediated systemic response to inflammation, similarly to other acute phase reactants [90].

About 80% of patients with early RA are positive for autoantibodies such as anti-citrullinated protein antibodies (ACPAs) and/or RF. These antibodies interact locally with antigens within joints, resulting in IC that trigger classical local complement activation. In addition, molecules released from the extracellular matrix of cartilage, including fibromodulin, osteomodulin, chondroadherin, the G3 domain of aggrecan and oligomeric matrix protein (COMP), and death cells debris, including extracellular DNA, are also direct triggers of complement activation [91,92,93,94].

The role of complement activation in RA is illustrated in models of collagen-induced arthritis (CIA), in which a compromised complement system led to a less severe disease [95,96,97,98] (Table 1). In this model, only the alternative pathway is essential for the development of the disease [99,100,101]. Factor H and its variant, factor H-1-like protein (FHL-1), may have a protective role against complement damage in the synovium [102], as has been also described in the collagen antibody induced arthritis (CAIA) model of arthritis [103] (Table 1). In addition, autoantibodies to factor H were found more frequently in RA patients compared to healthy controls [104]. Expression of CD55 in fibroblast-like synoviocytes is higher compared to leukocytes and endothelial cells [105]. In addition to its regulator role in CS activation, CD55 can also act as a ligand for CD97 expressed in a variety of cells, including infiltrated macrophages and T lymphocytes, and therefore may contribute to ongoing inflammation of the joint through other mechanisms as well [106].

A more relevant model of RA is the KRN/I-Ag7 (or K/BxN) mouse model of arthritis, mediated by a polyclonal T-cell dependent autoantibody response to a ubiquitous cytoplasmic glucose-6-phosphate isomerase (GPI) [107] (Table 1). The KRN T cell receptor (TCR) recognizes a foreign antigen, bovine pancreas ribonuclease (RNase), in the context of major histocompatibility complex class II (MHC II) I-A^k^ [108]. Serendipitously, it also recognizes a ubiquitously expressed self protein, GPI, in the context of MHC II I-Ag^7^ [109,110]. The KRN/I-A^g7^ mouse develops an inflammatory joint disease with many characteristics of RA. A key feature of KRN/I-A^g7^ model is that the arthritis can be transferred by serum or with monoclonal antibodies to a wide range of mouse strains [111,112]. Transfer of these autoantibodies are sufficient to mediate destructive arthritis, and represent a good model for ACPA positive RA as non-tissue specific autoantibodies-mediated arthritis model. In arthritic mice, these GPI deposits localized with IgG and C3 complement. Similar deposits were found in human arthritic joints, suggesting that GPI-anti-GPI complexes on articular surfaces initiate an inflammatory cascade via the alternative complement pathway, which is uncontrolled because the cartilage surface lacks the usual cellular inhibitors [113]. In the same sense, circulating C3 appears as an element necessary and sufficient for arthritis induction in this model [114]. However, other authors have shown that the presence of C3 is not necessary for the development of arthritis in this same model, suggesting that the pathogenesis can largely proceed by complement-independent pathways [115]. In addition, the classical and the lectin pathways are not involved in this model of antibody-mediated arthritis induction [116]. In contrast, in the absence of factor B, most of the animals did not develop arthritis [99], and similarly, oral administration of a factor B inhibitor prevented KRN-induced arthritis in mice [117].

The C5a-C5aR axis is important in the onset of inflammation in RA synovium and this pathway may be a relevant target for treatment of these patients [118]. Other therapeutic approaches with complement inhibitors have been tested in animal models of the disease in a non-specific way (e.g., with VSIG4, an Ig superfamily complement receptor, in CIA and CAIA or in a targeted way e.g., against Cr2-factor H complex in CAIA). Most of these inhibitors act at the level of C3 activation. In the animal models, different molecules have also been tested to inhibit C5, administered systemically or at a local intra-articular level, with good results.

## 6. ANCA Associated Vasculitis

The CS has a well-established pathogenic role in ANCA associated vasculitis (AAV) [119] and in the endothelial damage in large vessel vasculitis such as Takayasu arteritis [120]. Homozygous mutations of factor I had been identified as a rare monogenic cause of IC associated small vessel vasculitis [121]. These patients show a complete absence of alternative pathway activity, and a decline in the classical pathway with decreased serum levels of factor I, C3, and factor H but normal levels of C4. The first evidence of the pathogenic role of complement activation in AAV was provided by the mouse model of anti-myeloperoxidase (MPO) antibodies-induced vasculitis [122,123] (Table 1). C4-deficient mice are not protected against vasculitis induced by transfer of anti-MPO antibodies, whereas C5 deficiency confers protection [122]. Treatment with complement depleting agents slows down the development of anti-MPO antibody-induced renal involvement [122]. In this model, C4-deficiency, which precludes complement activation by the classical and lectin pathways, does not prevent the development of glomerulonephritis and vasculitis, whereas animals with a deficient alternative pathway were protected [122]. The blockade or deficiency of the C5a receptor (C5aR) also protects mice humanized with human C5aR against ANCA-induced glomerulonephritis [124]. In this study, C5aR deficiency improved the effect of induced glomerulonephritis, while animals with human C5aR showed a disease similar to that of wild type mice. Therapy with the human C5aR inhibitor CCX168 was used to block human C5aR in mice, resulting in a drastic decrease of necrosis and crescents in the glomeruli and less neutrophil infiltration in the glomeruli [124].

Neutrophils activated by the presence of ANCA release factors such as properdine, factor B, proteases, ROS, and MPO that can directly damage the endothelium and activate the alternative pathway generating in turn C5a with a high chemoattracting capacity on neutrophils [125]. Therefore, CS activation enhances the effect of autoantibodies on neutrophils, resulting in severe necrotizing inflammation of the vessel wall. It remains unclear how the alternative pathway is activated but recently, it was suggested that it might be activated by activated platelets [126]. As the main regulator of the alternative pathway, factor H may play an important role in limiting the secondary activation of complement. In fact, plasma levels of factor H were significantly lower in patients with active AAV compared to patients with AAV in remission and healthy controls [127]. Further research has shown that factor H in AAV patients is generally less active in binding and regulating C3b and in protecting cells from complement damage [128]. Clinical data show that in AAV patients, plasma and urinary complement fragments levels increase during disease flares [129,130]. Histological observations also confirm the presence of renal IC and C3c, C3d, C4d, and C5-9 deposition in correlation with proteinuria and renal failure [125,131].

## 7. Other

There is no definitive evidence for the pathogenic involvement of the complement system in systemic sclerosis. However, the presence of hypocomplementemia is clinically associated with characteristic features of overlap syndromes such as vasculitis or inflammatory myopathy [132]. Eculizumab has been anecdotally used in patients with scleroderma renal crisis with improved renal function and hypertension [133].

Complement-mediated damage has been studied predominantly in DM [134]. Autoantibodies and the CS mediate the immune damage manifested by the deposition of C3b, C4b and MAC in endomysial capillaries in the early stages of the disease [135]. Similarly, affected cutaneous areas show MAC deposits, while unaffected skin does not [136]. Somewhat paradoxically, genetic deficiency of C2 and C9 has been described associated with DM [135,137].

## 8. The Complement System as a Therapeutic Target

There are currently two CS blockers approved for therapeutic use, eculizumab, and plasma C1 protease inhibitor (C1INH). Eculizumab was approved for the treatment of paroxysmal nocturnal hemoglobinuria (PNH), atypical hemolytic uremic syndrome (aHUS) [138,139,140] while C1INH is indicated for the treatment of hereditary angioedema [141]. Eculizumab has also been approved for the treatment of acetylcholine receptor antibody positive refractory MG [142] and aquaporin antibody-positive (AQP)-4 neuromyelitis spectrum disorder [143].

The development of CS inhibitors is a difficult task. The target proteins are present in a high concentration in the circulation, with a rapid turnover and therefore, high doses of inhibitors or very frequent administration of antagonists seems necessary. Moreover, some redundancy in the activation pathways may limit the final effect of antagonists of upstream factors. The physiological roles of CS in IC processing, defense to intracellular bacteria, and potentially cell regeneration and repair cycles predict some potential side-effects of CS antagonists. The phenotype of patients with homozygous complement deficiencies include autoimmune disease, enhanced susceptibility to infections by Neisseria [144,145,146].

Different strategies to target CS are being developed that include the use of natural blockers such as factors H and I, pathogen-derived inhibitors, neutralizing antibodies or antibody fragments, peptides, small molecules, aptamers, siRNA, antisense oligonucleotides, and small molecules [147].

Eculizumab is the first approved CS antagonist. It is a designed IgG-kappa humanized monoclonal antibody, with an IgG2/IgG4 hybrid Fc component. Eculizumab specifically binds to C5 protein and inhibits its cleavage, thus preventing the generation of active C5a and C5b, and the C5b-C9 [148]. Although initially developed to treat autoimmune diseases such as RA, SLE, and DM [148] (and pexelizumab, a single-chain version was studied in acute myocardial infarction [149,150]), it finally obtained approval for the treatment of PNH. PNH is a clonal hematopoietic stem cell disorder that manifests with hemolytic anemia, bone marrow failure, and thrombosis as result of an acquired deficiency of complement regulatory proteins [151]. Following the results of a pilot study [152], two landmark trials confirmed its remarkable therapeutic efficacy. A randomized, placebo-controlled trial in transfusion-dependent patients, showed reductions in transfusion dependence and intravascular hemolysis, and an increased quality of life [140,153]. In a further open-label trial [153,154], these findings were confirmed in a larger patient population. Post-marketing and observational data [139,154] also showed a reduced risk of thrombosis [155,156]. Hemolytic uremic syndrome (HUS) is a thrombotic microangiopathy characterized by intravascular hemolysis, thrombocytopenia, and acute kidney failure. Atypical HUS (aHUS) is usually caused by uncontrolled complement activation, or as secondary HUS with a coexisting disease [157]. In 2013, Legendre et al. published two phase II trials of eculizumab in aHUS; one cohort with active thrombotic microangiopathy (TMA) and thrombocytopenia that did not respond to plasma treatment, and the other included patients with chronic plasma-dependent disease. Eculizumab improved the event-free status of TMA, increased glomerular filtration rate, allowing discontinuation of dialysis, and improved QoL [138], and these results persisted years after the end of the trial [158]. The efficacy of eculizumab in aHUS was then confirmed in two phase III trials in adults [159] and children [160]. The efficacy of eculizumab in other CS-mediated glomerular diseases such as C3 glomerulopathy has also been suggested by different case reports [161].

Ravulizumab (ALXN1210), a humanized monoclonal antibody to complement component C5, was engineered from eculizumab to have a substantially longer terminal half-life, permitting longer dosing intervals for PNH treatment, and achieving noninferiority for all efficacy and safety endpoints [162,163]. Similarly, ravulizumab has efficiently been used in adult patients with aHUS [164].

In view of the relevant role of CS in SLE, its therapeutic blockade seems a rational goal (Table 2). However, complement blockade therapy has only sporadically been tested in SLE. Genetic deficiencies of the components of the classical pathway may be successfully overcome by repeated plasma infusions, which result in disappearance of SLE symptoms [165]. Based on the effects of long-term treatment with anti-C5 on lupus nephritis in NZB/W lupus mice [166], a phase I study with eculizumab was conducted in 24 patients with SLE. In this single-center, randomized, placebo-controlled, double-blind trial with a range of doses, patients received a single intravenous dose of eculizumab or placebo and were followed for two months. Only mild adverse events were reported and a 10-day inhibition of the complement was observed in the 8 mg/kg group [167]. Eculizumab has been used off-label in patients with proliferative lupus nephritis and may be effective in patients with severe disease, but further randomized trials are clearly needed [168,169].

Several case reports also describe the use of eculizumab in SLE patients with TMA [167]. In patients with SLE and aHUS treated with eculizumab, a good tolerance and improvement of clinical, hematologic, and renal manifestations have been reported [169,170,171,172,173].

Current therapy for APS is based on anticoagulation. In catastrophic APS (CAPS), the most difficult to treat syndrome in which patients have multiple organ thrombosis, eculizumab has been used in patients with diseases refractory to anticoagulation or conventional immunosuppression getting a sustained remission [174].

In the case of RA, different strategies have been tested in animal models with complement blockers such as soluble CR1 (which suppresses complement activation at the C3 level) [175], or recombinant C3a and C5a receptor antagonists [176,177], and CD59 (which inhibits the formation of MAC) [178]. Selective intrarticular administration of an anti-C5 antibody showed therapeutic potential in the prevention or treatment of arthritis in a preclinical animal model for RA [179,180]. However, in humans, C5a blocking with an oral C5aR inhibitor PMX53 has not shown promising results [181] (Table 2). Neither has treatment with eculizumab showed efficacy in RA [182].

In 2000, the FDA granted eculizumab orphan drug status for the treatment of DM. This was followed by a pilot study to evaluate the safety and efficacy of the drug in 13 patients with the disease undergoing concomitant treatment with moderate doses of methotrexate or steroids. Although the results of the trial were positive in terms of safety and tolerability, the company did not publish any further information. A case report of a severe and refractory DM with TMA has also showed efficacy in this indication [183].

With respect to AAV, the rational points to the alternative pathway or the C5a-C5aR1 complex as therapeutic target. The first human studies are the CCX168 (avacopan) C5aR inhibitor trials (Table 2). This compound blocked the activation of C5aR after oral dosing in both humans and C5aR humanized mice (knock-in hC5aR mice). Recently, the results of phase I [184] and phase II [185] trials of CCX168 in patients with a clinical diagnosis of granulomatosis with polyangeitis, microscopic polyangeitis or vasculitis limited to the kidney have been published. In the phase I human clinical study, CCX168 was well tolerated, with excellent oral bioavailability and proportional dose increases in systemic exposure. Analysis of human data revealed that 30 mg of CCX168 twice daily provided excellent C5aR coverage in human blood leukocytes. The CLEAR (C5aR inhibitor in leukocyte-associated exploratory renal vasculitis) study was designed to assess whether the high doses of chronic steroids used in the standard of care (SOC) regimen for AAV could be reduced or eliminated—without compromising efficacy—by substitution with CCX168. The trial met its primary endpoint based on the Birmingham Vasculitis Activity Score (VAS) response. The proportion of patients with a VAS response at week 12 was numerically higher and statistically not lower among patients receiving CCX168 plus low-dose steroids and no steroids compared to the high-dose steroid-treated group A higher proportion of patients receiving CX168 had early clinical remission at week 4, sustained until week 12. Patients receiving CX168 (especially without steroids) showed significant improvement in health-related quality of life domains such as physical functioning, mental health, emotional well-being, pain, and vitality. Glomerular filtration rates improved similarly with CCX168 compared to controls, indicating that steroids are not needed to improve kidney function when patients are taking CCX168. Recently, results from the phase 3 clinical trial were announced. This study aimed to evaluate the safety of avacopan for the treatment of AAV in addition to SOC treatment with glucocorticoids with cyclophosphamide or rituximab. Although pending final publication, the preliminary results are very positive. [186].

There is a wide current development of blocking agents against different components of complement such as APL-2 (pegcetacoplan derivative of compstatin that blocks C3) or narsoplimab (that blocks MASP2) in lupus nephritis, and IFX-1 (that blocks C5a) in AAV [187].

Due to its multiple anti-inflammatory and immunomodulatory properties, IVIG is successfully used in a wide range of autoimmune and inflammatory conditions. IVIG has outstanding properties not only in neutralizing the activity of autoantibodies and suppressing the production of autoantibodies, but also in modulating the activity of CS. IVIG can exert its effects by several mechanisms: (a) linking the activated C3 and C4 and avoiding the deposition in situ of these fragments, (b) linking C1q and causing a deviation of C1 from its target, (c) enhancing the inactivation of C3 in the complex with immunoglobulins and thus regulating downwards the activity of C3 convertase, (d) modulating a mild and controlled activation of CS [188].

Practical experience has been gained in the area of infection risk mitigation through the clinical use of eculizumab and the means by which Neisseria infection has been partially mitigated through long-term antibiotic treatment and/or vaccination for Neisseria a month before starting therapy with eculizumab [148].

## 9. Conclusions

The pathogenic involvement of the CS in systemic autoimmune diseases has been described both by the genetic or functional deficiencies of certain components and by an excessive activation of the CS in both preclinical and human models. This has provided the pathogenic basis for the development of therapies aimed at blocking certain facets of complement. However, this development is still limited and, therefore, additional randomized and controlled studies are needed to confirm the usefulness of these directed therapies.

## Figures and Tables

**Figure 1 cells-10-00148-f001:**
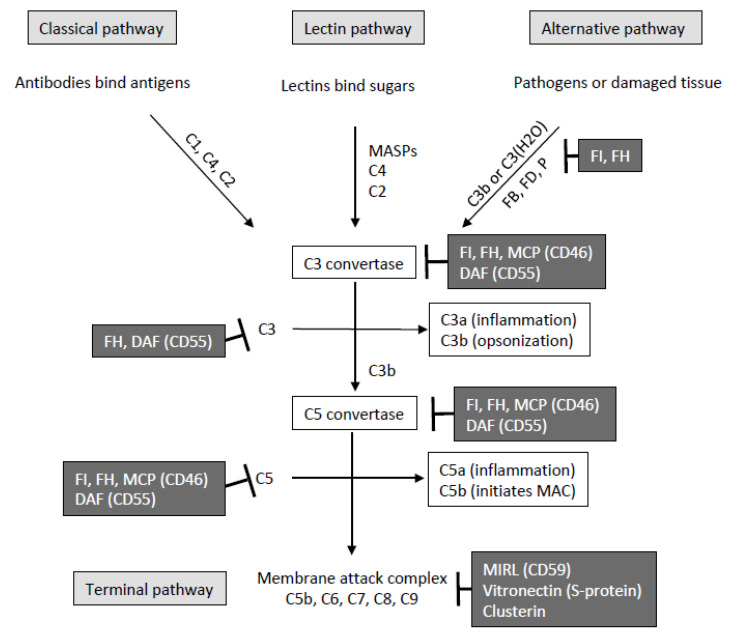
Complement system. There are three activation pathways in the complement system: classical, lectin, and alternative. All three pathways lead to the formation of C3 and C5 convertases, which rapidly amplify the complement response. In addition to the processes described above, several complement regulatory proteins are able to inhibit complement by inactivation of C3 and C5, and C3 and C5 convertases, or by preventing successful formation of the membrane attack complex. DAF: decay-accelerating factor or CD55; FB: factor B; FD: factor D; FH: factor H; FI: factor I; MASPs: MASP: MBL-associated serine proteases; MCP: membrane cofactor protein or CD46; MIRL: membrane inhibitor of reactive lysis or CD59; P: properdin.

**Table 1 cells-10-00148-t001:** Complement deregulation in autoimmune animal models.

Model Disease	Complement Deregulation	Therapeutic Management	Clinical Consequences
Antiphospholipid syndrome	C3 or C5 deficiency	C5b-9 blockade Factor B blockade C5 blockade	No fetal losses No thrombosis No fetal losses No fetal losses No fetal losses, no thrombosis
Rheumatoid arthritis CIA/CAIA CIA/CAIA KRN/I-Ag7 KRN/I-Ag7 KRN/I-Ag7	Alternative pathway Alternative pathway Factor B absence	C3 or C5 blockade Factor B blockade	Arthritis development No arthritis development Arthritis development No arthritis development No arthritis development
AAV Anti-MPO induced vasculitis	C5 deficiency C5aR deficiency	C5aR blockade	No vasculitis development No vasculitis development No vasculitis development

AAV: ANCA associated vasculitis; CAIA: collagen antibody induced arthritis; CIA: collagen induced arthritis; KRN/I-Ag7 (or K/BxN) mouse model of arthritis, mediated by a polyclonal T-cell dependent autoantibody response to a ubiquitous cytoplasmic glucose-6-phosphate isomerase.

**Table 2 cells-10-00148-t002:** Clinical studies of drugs targeting complement pathways in rheumatic autoimmune diseases.

Agent	Disease	Type of Study	Therapeutic Effect
Eculizumab (C5a inhibitor)	SLE SLE and TMA CAPS RA DM	Clinical Trial (phase I) Case Report Case Report Clinical Trial (phase IIb) Clinical Trial (placebo-controlled double blind pilot study) Case Report	No evidence Improvement Improvement No evidence No evidence Improvement
PMX53 (C5aR inhibitor)	RA	Clinical trial (placebo-controlled double blind pilot study)	No evidence
CCX168 (C5aR inhibitor)	AAV	Clinical Trial (phase II, III)	Improvement

AAV: ANCA associated vasculitis; CAPS: catastrophic antiphospholipid syndrome; DM: dermatomyositis; RA: rheumatoid arthritis; SLE: systemic lupus erythematosus; TMA: thrombotic microangiopathy.

## Data Availability

Not applicable.

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
