# Peer review of "Complement as a Therapeutic Target in Systemic Autoimmune Diseases"

_cells, 2021, doi:10.3390/cells10010148_

Round 1
Reviewer 1 Report
In this review, Izquierdo and colleagues describe the relation between the deregulation of complement homeostasis and the chronic inflammatory disorders. Although this review is in general sound and well written, I would recommend the authors to dramatically improve the description of the three different complement pathways (for beotian readers) and describe how anaphylatoxins trigger inflammation - describe ligand/receptor interaction (cells expressing these receptors) and cell signaling (brief description).
Also, could the authors present a table with the link between complement deregulation and the SLE (other)-like animal models?
Could the authors report the definition of each acronyme and define the genetic background of KRN mice?
Also I did not get access to figure 1, could the authors fix this issue ?
Author Response
Reviewer #1:
Comments to author
Q1: In this review, Izquierdo and colleagues describe the relation between the deregulation of complement homeostasis and the chronic inflammatory disorders. Although this review is in general sound and well written, I would recommend the authors to dramatically improve the description of the three different complement pathways (for beotian readers) and describe how anaphylatoxins trigger inflammation - describe ligand/receptor interaction (cells expressing these receptors) and cell signaling (brief description).
We greatly appreciate the comments of reviewer 1. In accordance with the reviewer's recommendations, the description of the three ways of activating the complement system has been expanded and relevant references have been included (page 2, lines 53-60).
Similarly, the involvement of anaphylotoxins in triggering the inflammatory response has been slightly expanded (page 3, lines 73-82).
Q2: Also, could the authors present a table with the link between complement deregulation and the SLE (other)-like animal models?
According to the reviewer's suggestion, a new table has been included (new table 1). Page 6, line 193; page 8, lines 250-251, 254, 263; page 9, line 299.
Q3: Could the authors report the definition of each acronyme and define the genetic background of KRN mice?
We regret the cases of acronyms where we have forgotten to include the definition. These mistakes have been solved (page 2, lines 63-64; page 3, line 96; page 8, lines 253-254). Similarly, we have expanded the description of the genetic background of KRN mice (page 8, lines 261-269).
Q4: Also I did not get access to figure 1, could the authors fix this issue?
We regret not having correctly attached the pdf file with the figure and tables of the article. We will attach it again, having included a new table with the link between complement deregulation and the animal models.
Reviewer 2 Report
Critique
- On page 3, line 19 please indicate the type of proliferative nephritis you are referring to.
- On page 4, line 7, C5b-C9 complex needs to be replaced with C5b-9 complex.
- Page 3 line 10, sC5b9 should be replaced with sC5b-9.
- On page 5, line 11 : Please explain what "complete absence of the alternative pathway activity means? Activation and consumption?
- Page 6 under The complement system as therapeutic target please add that Eculizumab has been approved also for treatment of MG and NMOSD.
- Last sentence of the manuscript please add that vaccination for Neisseria is needed a month before starting therapy with eculizumab.
- The manuscript needs a conclusion. As it is now it ends abruptly without any conclusions.
- Very few references are introduced into the manuscript in Introduction were complement activation pathways are presented. This need to be corrected.
- Figure 1 is missing
- A new Figure or Table is needed to present all the current therapeutic agents (approved or under exploration), their mechanism of action and diseases in which they were investigated.
Author Response
Reviewer #2:
Comments to author
Q1: On page 3, line 19 please indicate the type of proliferative nephritis you are referring to.
We greatly appreciate the comments of reviewer 2. According to the reviewer's comments, it is true that in the referenced work, the histological type of lupus nephritis is not specified and it is more correct to talk about active kidney involvement according to the activity score described by the authors. For this reason, the term "proliferative" has been changed by "active" (page 5, line 156).
Q2: On page 4, line 7, C5b-C9 complex needs to be replaced with C5b-9 complex.
According to the reviewer's suggestion, the mistake has been solved (page 6, line 197).
Q3: Page 3 line 10, sC5b9 should be replaced with sC5b-9.
According to the reviewer's suggestion, the mistake has been solved (page 6, line 200).
Q4: On page 5, line 11: Please explain what "complete absence of the alternative pathway activity means? Activation and consumption?
According to the reviewer’ comments, the genetic deficiency of factor I is a very rare disease and it is not clear what is the pathogenic effect. In some patients it may provoke an exaggerated risk of recurrent infections, while and in other patients it has been related to the development of autoimmunity.
Q5: Page 6 under The complement system as therapeutic target please add that Eculizumab has been approved also for treatment of MG and NMOSD.
According to the reviewer’ comments, this sentence has been included with relevant references (page 10, lines 343-345). And, therefore, lines 383-387 in page 12 have now been deleted.
Q6: Last sentence of the manuscript please add that vaccination for Neisseria is needed a month before starting therapy with eculizumab.
According to the reviewer’ comments, this sentence has been included (page 14, line 460-461).
Q7: The manuscript needs a conclusion. As it is now it ends abruptly without any conclusions.
According to the reviewer’s suggestion a Conclusions paragraph has been included (page 14, lines 463-469).
Q8: Very few references are introduced into the manuscript in Introduction were complement activation pathways are presented. This need to be corrected.
According to the reviewer’ comments, references have been added in Introduction (page 2, lines 53, 55, 63, 64, 66; page 3, lines 73, 82, 90, 93, 96, 102, 103; page 4, line 107).
Q9 and Q10: Figure 1 is missing. A new Figure or Table is needed to present all the current therapeutic agents (approved or under exploration), their mechanism of action and diseases in which they were investigated.
We regret not having correctly attached the pdf file with the figure and tables of the article. We will attach it again, having included a new table with the link between complement deregulation and the animal models.
Reviewer 3 Report
This is a relatively brief, but well-constructed, review of the roles of complement in various complement-mediated autoimmune diseases. This serves as a good introduction for those new to complement as it nicely frames what is known and what remains unresolved.
Of note, figure 1 was not attached for review unfortunately.
There are several minor issues to address:
1) There are minor but numerous grammatical errors, which is distracting. Review by native English speaker is recommended.
2) The introduction is a bit under-referenced: i.e. quantity of complement proteins being higher in the brain and kidney, C4a as ligand for PARs, range of cells that express complement receptors. References to other reviews is sufficient.
3) In the SLE section: C1q association with NPSLE is overstated. Authors reference a case report (van Schaarenburg et al), which in turn, references a case series (Kamphylafka et al) where van Schaarenburg et al states in the discussion that C1q-deficiency has a higher prevalence of NPSLE than complement-sufficient patients (20% vs <5%). Nowhere in the Kamphylafka et al study is this data presented. While observations do exist demonstrating a potential role of C1q in neuronal health (which these authors appropriately reference), this is not at all a well-established etiology for NPSLE. I urge the authors to substantially tone down the association or even remove it from the review.
4) Also in the SLE section: A couple important references that first demonstrated a relationship between complement and SLE are missing: Vaughan et al, J Lab Clin Med, 37:698, 1951; and Schur & Sandson, NEJM, 278:533, 1968.
5) In the therapy section: ravulizumab is also available for use. A paragraph or two about it and differences from eculizumab should be included.
Author Response
Reviewer #3:
Comments to author
This is a relatively brief, but well-constructed, review of the roles of complement in various complement-mediated autoimmune diseases. This serves as a good introduction for those new to complement as it nicely frames what is known and what remains unresolved.
Q1: Of note, figure 1 was not attached for review unfortunately.
We regret not having correctly attached the pdf file with the figure and tables of the article. We will attach it again, having included a new table with the link between complement deregulation and the animal models.
There are several minor issues to address:
Q2: 1) There are minor but numerous grammatical errors, which is distracting. Review by native English speaker is recommended.
According to the reviewer’s recommendation, and after having reviewed the manuscript again, we have corrected the grammatical errors we have detected.
Q3: 2) The introduction is a bit under-referenced: i.e. quantity of complement proteins being higher in the brain and kidney, C4a as ligand for PARs, range of cells that express complement receptors. References to other reviews is sufficient.
According to the reviewer’ comments, references have been added in Introduction (page 2, lines 53, 55, 63, 64, 66; page 3, lines 73, 82, 90, 93, 96, 102, 103; page 4, line 107).
Q4: 3) In the SLE section: C1q association with NPSLE is overstated. Authors reference a case report (van Schaarenburg et al), which in turn, references a case series (Kamphylafka et al) where van Schaarenburg et al states in the discussion that C1q-deficiency has a higher prevalence of NPSLE than complement-sufficient patients (20% vs <5%). Nowhere in the Kamphylafka et al study is this data presented. While observations do exist demonstrating a potential role of C1q in neuronal health (which these authors appropriately reference), this is not at all a well-established etiology for NPSLE. I urge the authors to substantially tone down the association or even remove it from the review.
We appreciate the reviewer's comment with which we fully agree. After reviewing again what has been published in the literature, it is true that the suspicion of this association is based on clinical cases without the pathogenic mechanism being completely defined. For this reason, we have decided to withdraw the sentence: “Neurological involvement is more frequent in patients with C1q deficiency, possibly due to the role of C1q in neuronal interactions” and the reference (page 4, line 128).
Q5: 4) Also in the SLE section: A couple important references that first demonstrated a relationship between complement and SLE are missing: Vaughan et al, J Lab Clin Med, 37:698, 1951; and Schur & Sandson, NEJM, 278:533, 1968.
According to the reviewer’ comments, references have been added (page 5, line 154).
Q6: 5) In the therapy section: ravulizumab is also available for use. A paragraph or two about it and differences from eculizumab should be included.
According to the reviewer’ comments, references have been added (page 12, lines 383-387).
Round 2
Reviewer 2 Report
Critique
1. In Figure 1 some of the abbreviations are not shown in the text of the manuscript.
-For instance THBD which I suppose is an abbreviation for Thrombospondin?
-What about MIRL? MIRL is now called CD59 and this correct name should be introduced in the Figure
2. In the same Figure the authors show that Factor I, Factor H, MCP, DAF, THBP inhibit C3a, C3 or C5, C5a is incorrect. See explanation bellow:
"The complement system is regulated at several levels: CD55, CR1, CD46, C4bp, and factors I and H regulate the activity of the C3 convertase and C5 convertase, and other proteins such as CD59 block the final assembly of the pores by preventing the binding of C9. The S protein/vitronectin binds to C5b-7 and leads to the formation of a cytolytically inactive SC5b-9 complex"
Please modify the Figure accordingly.
3. In addition please delete THBD as this is not recognized as a classical inhibitor of these convertases.
4. Please introduce S-protein as this protein is an important inhibitor of lytic MAC formation
Author Response
Reviewer #2:
Comments to author
Q1. In Figure 1 some of the abbreviations are not shown in the text of the manuscript.
-For instance THBD which I suppose is an abbreviation for Thrombospondin?
-What about MIRL? MIRL is now called CD59 and this correct name should be introduced in the Figure.
We greatly appreciate the comments of reviewer 2. In accordance with the reviewer's recommendations, abbreviations have been added at the bottom of the figure.
TBHD corresponds to thrombomodulin and provide cofactor activity for factor I.
- In the same Figure the authors show that Factor I, Factor H, MCP, DAF, THBP inhibit C3a, C3 or C5, C5a is incorrect. See explanation bellow:
"The complement system is regulated at several levels: CD55, CR1, CD46, C4bp, and factors I and H regulate the activity of the C3 convertase and C5 convertase, and other proteins such as CD59 block the final assembly of the pores by preventing the binding of C9. The S protein/vitronectin binds to C5b-7 and leads to the formation of a cytolytically inactive SC5b-9 complex"
According to the reviewer's suggestion, mistakes have been corrected.
Please modify the Figure accordingly.
- In addition please delete THBD as this is not recognized as a classical inhibitor of these convertases.
According to the reviewer's suggestion, THBD has been deleted.
- Please introduce S-protein as this protein is an important inhibitor of lytic MAC formation.
According to the reviewer's suggestion, Vitronectin and clusterin have been added as inhibitors of lytic MAC formation.